# Health service research definition builder: An R Shiny application for exploring diagnosis codes associated with services reported in routinely collected health data

**Kelsey Chalmers**[1]*, **Valérie Gopinath**[1], **Adam G. Elshaug**[2]

**1** Lown Institute, Boston, Massachusetts, United States of America, **2** Melbourne School of Population and Global Health, The University of Melbourne, Melbourne, Victoria, Australia

* kchalmers@lowninstitute.org

**Data Availability Statement:** The claims data cannot be shared publicly because researcher and data access needs to be approved by the Centers

## Abstract

Many administrative health data-based studies define patient cohorts using procedure and diagnosis codes. The impact these criteria have on a study's final cohort is not always transparent to co-investigators or other audiences if access to the research data is restricted. We developed a SAS and R Shiny interactive research support tool which generates and displays the diagnosis code summaries associated with a selected medical service or procedure. This allows non-analyst users to interrogate claims data and groupings of reported diagnosis codes. The SAS program uses a tree classifier to find associated diagnosis codes with the service claims compared against a matched, random sample of claims without the service. Claims are grouped based on the overlap of these associated diagnosis codes. The Health Services Research (HSR) Definition Builder Shiny application uses this input to create interactive table and graphics, which updates estimated claim counts of the selected service as users select inclusion and exclusion criteria. This tool can help researchers develop preliminary and shareable definitions for cohorts for administrative health data research. It allows an additional validation step of examining frequency of all diagnosis codes associated with a service, reducing the risk of incorrect included or omitted codes from the final definition. In our results, we explore use of the application on three example services in 2016 US Medicare claims for patients aged over 65: knee arthroscopy, spinal fusion procedures and urinalysis. Readers can access the application at https://kelsey209.shinyapps.io/hsrdefbuilder/ and the code at https://github.com/kelsey209/hsrdefbuilder.

## Introduction

An estimated 76% of studies using administrative health databases use diagnosis or procedure codes to define patient cohorts, exposures or outcomes [1]. This includes those mapping clinical guidelines and/or recommendations to claims-based indicators to measure low-value care, including our own previous work [2–4]. Developing these definitions can be a lengthy and

of Medicaid and Medicare (CMS) Research Data Assistance Center. IRB restrictions and CMS approvals are required for access to the CMS data. Details on accessing the data can be found here: https://resdac.org/, or researchers can reach out to resdac@umn.edu. The aggregated data underlying the results presented in the study and all code to generate the results are available from https://github.com/kelsey209/hsrdefbuilder.

**Funding:** The authors received funding from Arnold Ventures LLC for this project (ID 2004079) (https://www.arnoldventures.org/). The funders had no role in study design, data collection and analysis, decision to publish, or preparation of the manuscript.

**Competing interests:** I have read the journal's policy and the authors of this manuscript have the following competing interests: AE reports receiving personal fees from the Australian state government health departments – Victoria, Queensland, South Australia, as well as the Australian Department of Veterans Affairs, Medibank Ltd, Private Healthcare Australia, and the Australian Defence Force Joint Health Command and grants from the National Health and Medical Research Council outside the submitted work. AE is also on the board of the New South Wales Bureau of Health Information. This does not alter our adherence to PLOS ONE policies on sharing data and materials.

exhaustive process, particularly in research work where data access is restricted to one or a few analysts. Mapping requires code selection from a codebook or dictionary, clinical input and face validity checks, and clinical coder input. Even with the last two steps the risk of inaccuracies may be high. Researchers working on mapping diagnosis codes from a guideline or recommendation may not have a complete view of all necessary criteria, and clinical/coder input might also miss these in their input.

Here, we propose an approach which starts with a summarized output of claims data associated with the selected medical service or procedure and can then be used to guide collaborators' decisions through an interactive application, where users can select the inclusion/exclusion criteria and visualize the estimated cohort size. This also has the benefit of being able to share these summaries with external reviewers or groups, and researchers' can demonstrate the estimated impact of their decisions. This program and application can be applied under the following circumstances:

1. Researcher access to encounter-level claims data with a goal to develop a definition based on a group of patients receiving a recorded procedure or service,

2. Access to similar claims without the service, and

3. One or multiple diagnosis codes recorded with the claim (such as International Classification of Diseases [ICD] codes).

We developed a SAS program to find and summarize a sample of claims with a selected service, which readers can download and apply on their own data sets. This was developed using 2016 Medicare claims data accessed through the Centers for Medicare and Medicaid Services' (CMS) Virtual Research Data Center (VRDC), which only allows access to and analysis of claims data using SAS. The output of this can then be read by our R Shiny application; the Health Services Research (HSR) Definition Builder app. This allows users to select codes as inclusion or exclusion criteria and visualize the estimated impact this has on the final cohort counts. Readers can download the R package for this app https://github.com/kelsey209/hsrdefbuilder, or they can access an online version here https://kelsey209.shinyapps.io/hsrdefbuilder/.

Briefly, the SAS program includes a tree classification model which predicts whether a service code was recorded in a claim from a random sample of claims with and without the service recorded. The diagnosis codes (or predictors in this model) with the highest relative importance values are used to iteratively group the service claims, essentially placing each claim in a 'branch' based on whether they had a particular diagnosis code or not. The SAS output is a table containing the counts of important diagnosis codes within each of these groups. The HSR Definition Builder uses this output to create an interactive table where users can select these codes to include or exclude from the total counts, and two graphs are updated to visualize these counts.

We use a selection of services here as a demonstration: knee arthroscopy, lumbar spinal surgery, and urinalysis (Table 1 shows the selected codes and definitions). Knee arthroscopy has been investigated in multiple claims-based studies on low-value care, as various studies have demonstrated its limited to no benefit compared to non-invasive treatments for certain patients [5,6]. Spinal fusion has also been included in multiple studies and has existing low-value care indicators [2,7,8]. Spinal fusion and arguably the other spinal procedures we included are defined as low-value or inappropriate for patients with low back pain [9–12]. Since low back pain can relate to multiple potential diagnoses/labels, creating a complete definition can be challenging [13,14]. Finally, urinalysis is often cited and investigated as a potential low-value service in some circumstances, and some studies have investigated its use in

Table 1. The selected example codes and their definitions.

| Service | Code | Definition |
|---|---|---|
| Urinalysis | 81001 | Automated urinalysis using dip stick and microscopy of urine |
| | 81003 | Automated urinalysis using dip stick |
| Spinal procedures | 0SB0x | Excision of lumbar vertebral joint [any approach, device or qualifier] |
| | 0RG6x | Fusion of thoracic vertebral joint [any approach, device or qualifier] |
| | 0SG0x | Fusion of lumbar vertebral joint [any approach, device or qualifier] |
| Knee arthroscopy | 29877 | Surgical arthroscopy of knee with chondroplasty |
| | 29879 | Surgical arthroscopy of knee with abrasion arthroplasty |

claims data [15–17]. We include it as an example here to demonstrate the application's utility for investigating diagnosis codes associated with a more generally used pathology test.

## Methods and materials

### Data

We had access to the CMS Chronic Conditions Warehouse, including carrier, outpatient, and inpatient (and MedPar) claims. Our initial code selections and explorations were based on claims in calendar year 2016. In the selected service claims, we excluded patients that were younger than 65 on their date of service.

This study was deemed exempt from review and consent was waived by the WCG IRB (formerly New England IRB) as we used administrative claims data accessed through CMS ResDAC.

### Finding important diagnosis codes

We started with the premise that for a given service there exist some diagnosis codes relevant to forming part of some theoretical cohort definition (the *important* diagnosis codes). These codes may or may not be recorded as the principal diagnosis in the claims (that is, recorded as the main reason for the service) and could be secondary diagnoses. Some of these important codes may also be infrequently observed and/or recorded. A group of claims with the same service might have thousands of unique diagnosis codes–there are approximately 60,000 ICD-10-CM codes. Often the most frequent diagnosis codes recorded may be related to commonly occurring and recorded co-morbidities (for example, chronic hypertension).

In order to find these important diagnosis codes, we determined which codes were most unique to the service claims, regardless of frequency. We created a data set of claims with and without the service, and then used a classifier on this data set to find the diagnosis codes that are most predictive of whether a claim had the service or not. We used a classification tree with binary predictor variables for each diagnosis code in the input data (1 for present in claim, and 0 otherwise). The model iteratively splits and creates partitions of the data based on these predictor variables, until some stopping criteria is reached or all claims within these partitions (or 'leaves') have the same response value.

If a user was trying to build a predictive model to use on other data sets, using the full tree (that is, where the tree is grown until all leaves have the same response value) can overfit the training data meaning that the model does not generalize well to new data. To avoid overfitting, the final tree would be pruned so not all decision rules are included in the final model. In contrast, our goal was to allow largest possible tree depth, so that we could identify as many potential important diagnosis codes as possible. For example, we might find just 10 codes which accurately predict the service in 90% of claims, while 50 codes predict the service in 99%

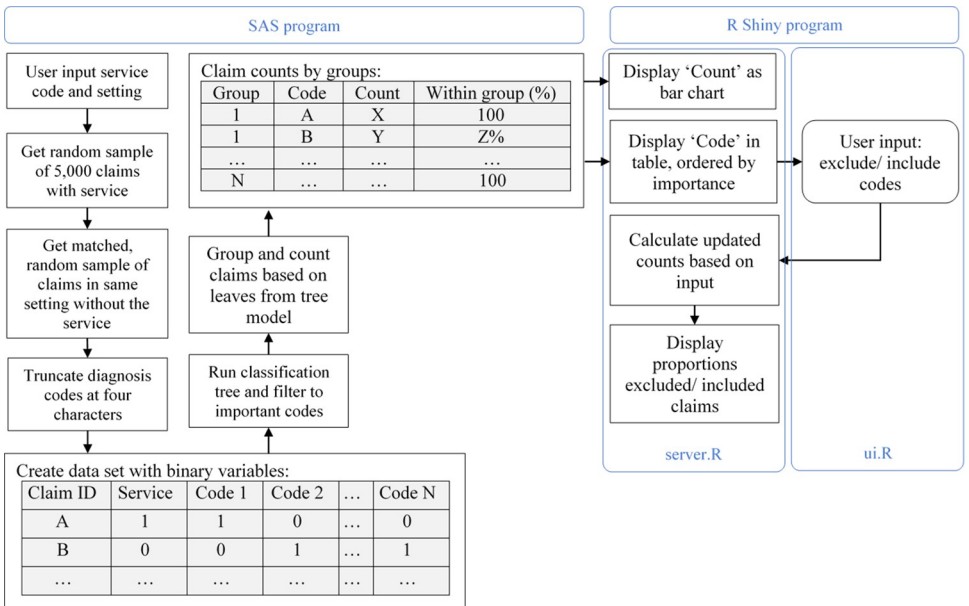

**Fig 1. The data processing steps in 1) SAS and 2) R Shiny (in the server and user-interface [ui] scripts).**

of claims. These additional 40 codes might be important edge cases for our cohort definition, and therefore we included them in our output.

Fig 1 shows the steps our SAS program uses to find these important diagnosis codes. First, a sample of 2016 claims with the input medical service or procedure is selected. The decision to use outpatient, carrier or MedPAR (inpatient) claims is based on the type of service and what the user wants to investigate. A matched random sample of claims in the same setting without the service is then selected. These claims are matched on patient age, sex, and the number of recorded diagnosis codes within the claim. This is to balance any diagnosis codes in the service and non-service set of claims that might be confounded with the age, sex or services for patients with higher rates of comorbidities recorded compared to other patients without the service.

We then created a data set combining the sample of service claims and the random sample of non-service claims, with a binary variable for each unique diagnosis code reported in the claims. Since this can create a dataset with thousands of variables, we decided to reduce the number of variables by truncating diagnosis codes at four characters.

We used the SAS function *hpsplit* to build a classification tree on this data set, using entropy as a measure to assess candidate splits of the data and a c45 pruning method for large enough trees [18]. We used the largest maximum depth value, 75, for the tree. We saved the diagnosis codes from this process that were identified as important features: those with a positive relative importance value.

We also created a filter for these features to select those diagnoses we expected to be more frequent in the service claims compared to the non-service claims. For each diagnosis code/ feature, we considered two proportions: the proportion of claims in which the code appears in the non-service sample claims, and the proportion of claims in which the code appears in the service claims. These proportions are random values, and if we re-sampled our full claims data we would have new estimates for the true proportion. We assumed that these had a *beta*-distribution, with parameters $\alpha = 1+p^*n$ and $\beta = 1+n-p^*n$, where $p$ is the proportion of claims with the code out of the total number of claims, $n$. This gives a wide confidence interval in cases

where the proportion is high; for example, if a code appeared in 50% of our 5,000 claims, the 95% confidence interval for the true proportion would be 0.49 to 0.51. If the proportion is low, then the width of the confidence interval is lower; for example, if a code appeared in 2% of our 5,000 claims, the 95% confidence interval would be 0.017 to 0.024.

We exclude important diagnosis codes with overlapping 95% confidence intervals of proportions in the service and non-service sample. Consider an example where we have 1,364 claims in the service and non-service groups, and there are 246 claims with a type 2 diabetes diagnosis code in the service group (18.0%) and 221 in the non-service group (16.2%). Using the *beta*-distribution described above, the 95% confidence interval for diabetes in the service group is 16.40 to 19.82%, while in the non-service group it is 14.64 to 17.92%. As these two intervals overlap, we would remove this type 2 diabetes from our list of important diagnosis codes.

## Creating summary of claims

Next, we generated output from SAS that can be exported from the VRDC and used in the R application. We grouped the service claims based on the remaining important diagnosis codes, using a similar concept as the tree classifier. In the first group, we included all claims that contained the diagnosis code with the highest relative importance value. In the second group, we included all remaining claims that contained the code with the second highest relative importance value. This process continued until all claims were in one group, or there were no remaining important diagnosis codes. For the output, we calculated the number and percentage of claims with each diagnosis code within each group and masked any counts that were less than eleven before downloading the table from the VRDC.

## R Shiny app to display these outputs

The R Shiny app displays three main components (Fig 2) [19]. One is the table of important diagnosis codes, ordered by decreasing relative importance values from the classification tree. The next is a bar graph that displays the relative claim counts in each group, with each group labelled with the relevant important diagnosis code. The third component is the single bar graph which shows the estimated claim number overall which meet the user's selected criteria. These graphics are created using the *ggplot2* [20].

A user can select diagnosis codes in the table and then choose to either 1) drop these from the table, 2) use them as an inclusion/exclusion criterion for a preliminary definition or 3) add a label to these codes.

As a user selects diagnosis codes to include or exclude from the counts, the two graphs are updated to reflect the claim counts matching the new definition. The total sum of the excluded and included claims with specific diagnosis codes are calculated from the SAS output table. If a user selects a diagnosis code which has a count of fewer than 11 claims (and is therefore masked in the SAS output table), then the application presents a range of potential counts. The maximum value presented is the sum of all claims with included diagnosis codes with any masked values replaced with 10, minus the sum of all excluded claims with any masked values replaced with 1. The minimum value shown is the sum of all included diagnosis codes with any masked values replaced with 1, minus the sum of all excluded claims with any masked values replaced with 10 (or, if this is negative, the minimum value is zero).

Finally, the application has an action button for users to download their preliminary definitions as comma-separated value files. It also has an action button for users to upload any previously downloaded definitions, allowing users to revise and share their definition with others.

**Fig 2. Two components of the *HSR Definition Builder* Shiny application, which allows users to explore diagnostic indications on claims data and run their own inclusion and exclusion criteria counts.** The example above shows our selected indications for 2016 outpatient claims for knee arthroscopy. *A*. The first panel lists the diagnoses codes associated with this procedure. Users have options to select these codes and put them as exclusion or inclusion criteria. Users can also select rows and create a label for multiple diagnosis codes. *B*. The estimated counts of claims that fall under the user's definition. The bar chart shows the claims with each diagnosis code. The donut chart shows the overall included, excluded or 'maybe' counts. The 'maybe' category exists due to the masking of small counts from the claims data.

## Results

Table 2 shows the number of 2016 Medicare claims for patients over 65 for each of our investigated services, and also reports the accuracy of the tree classification models. We opted to use a maximum sample of 5,000 claims to analyze and 1,000 claims as a validation set. For example, we randomly selected 6,000 of 5,505,088 carrier claims with urinalysis CPT code 81003.

**Table 2. Results for the tree classification models for our example services.**

| Service | Setting | Code type | Code | Claims (n) | Sensitivity (%) | Specificity (%) | Misclassification rate (%) | Groups (n) | Claims included (n, %) |
|---|---|---|---|---|---|---|---|---|---|
| Urinalysis | Carrier | CPT | 81001 | 6,900,907 | 93.44 [88.57] | 92.72 [77.75] | 6.92 [16.99] | 68 | 4,622 (92.44) |
| | | | 81003 | 5,505,088 | 88.22 [80.35] | 95.62 [81.87] | 8.08 [18.91] | 77 | 4,584 (91.68) |
| | Outpatient | CPT | 81001 | 6,218,336 | 79.86 [68.60] | 96.78 [80.49] | 11.68 [25.44] | 72 | 4,593 (91.86) |
| | | | 81003 | 2,601,114 | 78.48 [74.15] | 97.08 [66.64] | 12.22 [29.76] | 71 | 4,477 (89.54) |
| Spinal procedures | Inpatient | ICD | 0RG6 | 1,670 | 100 [93.03] | 100 [96.11] | 0 [5.44] | 35 | 1,669 (99.94) |
| | | | 0SB0 | 757 | 100 [96.92] | 100 [95.77] | 0 [3.68] | 15 | 754 (99.6) |
| | | | 0SG0 | 49,884 | 100 [98.50] | 100 [97.11] | 0 [2.21] | 19 | 5,000 (100) |
| Knee arthroscopy | Carrier | CPT | 29877 | 7,178 | 98.32 [96.16] | 98.96 [97.51] | 1.36 [3.14] | 51 | 4,931 (98.62) |
| | | | 29879 | 12,032 | 99.66 [98.90] | 99.60 [99.03] | 0.37 [1.03] | 29 | 4,971 (99.42) |
| | Outpatient | | 29877 | 1,364 | 99.78 [97.24] | 99.93 [96.54] | 0.15 [3.14] | 30 | 1,353 (99.19) |
| | | | 29879 | 3,105 | 100 [98.29] | 100 [98.89] | 0 [1.41] | 20 | 3,102 (99.90) |

The maximum number of included claims in the training data was set at 5,000, with 1,000 claims in the validation set. The sensitivity, specificity and misclassification rates are for the prediction (on the training data and in the validation data in the square brackets) of whether the procedure code was included in the claim based on tree classification. The claims included is the number of claims with at least one of the identified important diagnosis codes.

The sensitivity of our classification model was 88.22% (80.35% on the validation data), meaning the model correctly predict a urinalysis code in 4,411 out of 5,000 claims. The specificity was 95.62%; 4,781 out of 5,000 claims without a urinalysis code were correctly identified. This gives the model a misclassification rate of 8.08% (18.91% on the validation data). The 5,000 claims sample were divided into 77 distinct groups based on whether the claim had one of the important diagnosis codes or not. There were 4,584 claims (91.68%) that had at least one of these diagnosis codes.

The investigated procedure classifications ranged from 98.32 to 100% (96.16 to 98.90% for the validation claims) for knee arthroscopy and were 100% (93.03 to 98.50% for the validation claims) for spinal procedures. These are the sensitivity rates with specificity and misclassification also reported in Table 2.

The urinalysis classifications were lower than the two investigated procedures, ranging from 78.48 to 93.44%, and lower for the validation set of claims. This is not surprising given some of the selected important diagnosis codes for urinalysis are likely to be present in the claims without urinalysis, and patients who receive a routine urinalysis (meaning there is no indication) may look similar to other patients without a urinalysis.

## Demonstrating claim grouping: Knee arthroscopy example

Our first set of results demonstrates how the selected medical service or procedure claims are grouped by the important diagnosis codes. Table 3 shows the top 21 most important diagnosis codes for knee arthroscopy procedure (CPT 29877) in carrier claims (out of 132 total important diagnosis codes) based on relative importance in the classification model. Code S83.2 (tear of meniscus, current injury) had the highest relative importance, meaning it was most distinct to the service claims compared to a random set of claims.

**Table 3. Top 21 of 132 diagnosis codes for carrier claims with a knee arthroscopy procedure (CPT 29877), ordered by relative importance from the classification model.**

| Group | Code | Description | Total group count | Percentage in group (N = 5,000) | Total code count in sample |
|---|---|---|---|---|---|
| 1 | S832 | Tear of meniscus, current injury | 1906 | 38.12 | 1906 |
| 2 | M942 | Chondromalacia | 1012 | 20.24 | 1544 |
| 3 | M224 | Chondromalacia patellae | 578 | 11.56 | 1469 |
| 4 | M171 | Unilateral primary osteoarthritis of knee | 564 | 11.28 | 1299 |
| 5 | Z966 | Presence of orthopedic joint implants | 158 | 3.16 | 170 to 198 |
| 6 | M232 | Derangement of meniscus due to old tear or injury | 102 | 2.04 | 542 |
| 7 | M233 | Other meniscus derangements | 66 | 1.32 | 303 to 321 |
| 8 | M246 | Ankylosis of joint | 52 | 1.04 | 104 to 149 |
| 9 | M234 | Loose body in knee | 52 | 1.04 | 401 to 437 |
| 10 | M258 | Other specified joint disorders | 43 | 0.86 | 76 to 139 |
| 11 | M008 | Arthritis and polyarthritis due to other bacteria | 37 | 0.74 | 42 to 87 |
| 12 | M239 | Unspecified internal derangement of knee | 34 | 0.68 | 86 to 131 |
| 13 | M238 | Other internal derangements of knee | 25 | 0.50 | 59 to 122 |
| 14 | M948 | Other specified disorders of cartilage | 23 | 0.46 | 39 to 84 |
| 15 | M179 | Osteoarthritis of knee, unspecified | 26 | 0.52 | 72 to 135 |
| 16 | T848 | Other complications of internal orthopedic prosthetic device | 17 | 0.34 | 54 to 108 |
| 17 | S821 | Fracture of upper end of tibia | 18 | 0.36 | 23 to 68 |
| 18 | M009 | Pyogenic arthritis, unspecified | 17 | 0.34 | 22 to 67 |
| 19 | M255 | Pain in joint | 60 | 1.20 | 354 to 444 |
| 20 | M658 | Other synovitis and tenosynovitis | 13 | 0.26 | 222 to 312 |
| 21 | M125 | Traumatic arthropathy | 13 | 0.26 | 15 to 33 |

The first group in Table 3 contains all claims with diagnosis code S83.2 (38.12% of claims in the 5,000 sample). The next group contains the remaining claims all containing M94.2 chondromalacia), the code with the next highest relative importance. This is 1,012 (20.24%) of the 5,000 sample, and since the total code count for M942 is 1544, group 1 contains 532 claims with code M94.2.

Cell counts in the SAS output fewer than 11 are masked before download. The HSR Definition Builder calculates a range to represent the true claim count for these diagnosis codes, based on a range of 1 to 10 for each missing value. For example, there were 303 to 321 claims in the 5,000 sample with code M23.3 (other meniscus derangements). Based on the group order in Table 3, we see that 237 to 255 of these claims were in the previous groups (and therefore intersected with codes S83.2, M94.2, M22.4, M17.1 and/or Z96.6).

## Comparison to published definitions: Urinalysis example

N390, a urinary tract infection (UTI), was the most frequent important diagnosis code and the code with the highest relative importance for all urinalysis claims. Many of the important diagnosis codes are likely related to findings for the test, and not the indication for the test. A positive urinalysis for UTI may occur in up to 90% of asymptomatic elderly patients [17]. Likely indications for a urinalysis include R300 (dysuria), R350 (frequency of micturition), R109 (unspecified abdominal pain), R42 (dizziness and giddiness).

Table 4 compares the urinalysis algorithm from Shenoy et al [21], which counted low-value preprocedural urinalysis in a commercial and Medicare claims data set, to the same codes identified in the HSR Definition Builder. Our app selected six of the same codes in carrier claims and seven in outpatient claims (keeping in mind that our cohort included all urinalysis claims, opposed to preprocedural only). The codes from the Shenoy et al algorithm that were absent from the important diagnosis codes were the most infrequent in all urinalysis claims (Table 4). There were also codes identified in the app that may have been relevant for a low-value urinalysis definition. For example, N13.9 (obstructive and reflux uropathy, unspecified) was in the Shenoy et al algorithm, but not N13.8 (other obstructive and reflux uropathy) which was an important diagnosis code.

**Table 4. Comparing the Shenoy et al [21] algorithm for low-value urinalysis and important diagnosis codes in the *HSR Definition Builder* application.**

| Plausibly indicated urinalyses codes from the Shenoy et al algorithm | | Count per 1,000 claims | | | | HSR Definition Builder identified important diagnosis code | | | |
|---|---|---|---|---|---|---|---|---|---|
| | | Carrier, 81001 | Carrier, 81003 | Outpatient, 81001 | Outpatient, 81003 | Carrier, 81001 | Carrier, 81003 | Outpatient, 81001 | Outpatient, 81003 |
| N39 | Other disorders of the urinary system | 239.74 | 203.94 | 215.11 | 136.24 | X | X | X | X |
| R30 | Pain associated with micturition | 57.88 | 64.32 | 49.74 | 44.07 | X | X | X | X |
| R31 | Hematuria | 57.65 | 85.66 | 44.23 | 32.58 | X | X | X | X |
| N300 | Acute cystitis | 13.46 | 22.40 | 20.46 | 12.98 | X | X | X | X |
| R350 | Frequency of micturition | 46.10 | 74.11 | 30.95 | 34.09 | X | X | X | X |
| R3915 | Urgency of urination | 12.43 | 27.52 | 6.17 | 7.65 | X | X | X | X |
| R50 | Fever of other and unknown origin | 5.85 | 4.66 | 15.75 | 12.43 | | | X | X |
| N36 | Other disorders of the urethra | 1.16 | 2.18 | 0.57 | 0.75 | | | | |
| N34 | Urethritis and urethral syndrome | 0.91 | 1.36 | 0.65 | 0.53 | | | | |
| N139 | Obstructive and reflux uropathy, unspecified | 0.86 | 1.00 | 1.15 | 0.78 | | | | |
| R40 | Somnolence, stupor and coma | 0.68 | 0.28 | 2.87 | 2.59 | | | | |
| B088 | Other specified viral infections characterized by skin and mucous membrane lesions | 0.00 | 0.00 | 0.00 | 0.00 | | | | |

## Building a potential cohort definition: Spinal procedure example

Here, we use one of the spinal procedure outputs to develop a preliminary cohort definition for potential low-value care. We included inpatient admissions with a principal procedure code starting with 0SG0 (fusion of lumbar vertebral joint). There were 49,884 Medicare inpatient claims for this procedure in 2016. The classification model on the sample data had a 100% accuracy, and all of our sample (5,000 claims) had at least one of the important diagnosis codes.

M480 (spinal stenosis) was the diagnosis code with the highest relative importance. This code was in 3,787 (75.74%) of the 5,000 sample claims. Since spinal surgery is likely ineffective for spinal stenosis we did not exclude claims with this code [22]. Several codes with radiculopathy (M511, M541, M472) were also identified, and again we decided not to exclude these codes since there is a lack of evidence of long-term benefit [23]. G834 (cauda equina syndrome) was also one of the important diagnosis codes, which does require immediate surgery and therefore we selected this code as an exclusion criterion [24]. Note there were only 33 to 69 claims with this diagnosis code in the sample. Table 5 shows our final exclusion counts. Using this definition will identify and exclude 16 to 34% of claims for this spinal procedure as appropriate care.

**Table 5. Preliminary exclusion criteria for inpatient fusion of lumbar vertebral joint (ICD-10 codes 0SG0-x).**

| Exclusion label | Codes | Definitions | Claim count |
|---|---|---|---|
| Cancer | Z858 | Personal history of malignant neoplasms of organs and systems | 228 to 264 |
| | C795 | Secondary malignant neoplasm of bone and bone marrow | 5 to 50 |
| | M845 | Pathological fracture in neoplastic disease | 4 to 40 |
| | C794 | Secondary malignant neoplasm of and unspecified parts of nervous system | 1 to 10 |
| | D166 | Benign neoplasm of vertebral column | 1 to 10 |
| Cauda equina | G834 | Cauda equina syndrome | 33 to 69 |
| Discitis | M464 | Discitis, unspecified | 7 to 70 |
| | M463 | Infection of intervertebral disc (pyogenic) | 1 to 10 |
| Fracture | S320 | Fracture of lumbar vertebra | 22 to 67 |
| | M800 | Age-related osteoporosis with current pathological fracture | 4 to 40 |
| | M844 | Pathological fracture, not elsewhere classified | 3 to 30 |
| | M808 | Other osteoporosis with current pathological fracture | 3 to 30 |
| | S321 | Fracture of sacrum | 3 to 30 |
| | S220 | Fracture of thoracic vertebra | 3 to 30 |
| | M846 | Pathological fracture in other disease | 2 to 20 |
| Osteomyelitis | M462 | Osteomyelitis of vertebra | 9 to 90 |
| Scoliosis | M418 | Other forms of scoliosis | 183 to 255 |
| | M419 | Scoliosis, unspecified | 158 to 239 |
| | M412 | Other idiopathic scoliosis | 33 to 60 |
| | M415 | Other secondary scoliosis | 19 to 37 |
| | M411 | Juvenile and adolescent idiopathic scoliosis | 3 to 30 |
| | M413 | Thoracogenic scoliosis | 2 to 20 |
| Kyphosis | M402 | Other and unspecified kyphosis | 48 to 111 |
| | M963 | Postlaminectomy kyphosis | 4 to 40 |
| | M400 | Postural kyphosis | 2 to 20 |
| | M401 | Other secondary kyphosis | 3 to 30 |
| **Total excluded claim count** | | | 784 to 1702 (16 to 34%) |

The claim count ranges represent the total exclusions calculated in the HSR Definition Builder application out of 5,000 claims.

## Discussion

Our aim was to create and share a tool that will be useful for researchers developing cohort definitions; analysts can share insights into overlapping code counts, and these definitions (and what they miss) are transparent to reviewers and other audiences. Previous work has detailed methods of clustering and identifying clinically distinct cohorts in claims data; for recent examples see [25–27]. These techniques are useful to investigate co-occurring conditions and to visualize/label patient groups in particular health settings. Our approach is similar, but instead of aiming to identify and cluster different patient groups, we aimed to find a near complete set of diagnosis codes that describe all patients with the service. In addition, we provide an interactive tool which researchers can use to estimate cohort sizes depending on different combinations of diagnosis inclusion/ exclusion criteria whose selections they can control through the application tool. The results of each scenario can be generated quickly and easily shared. With clustering, comorbidities in particular may overlap different distinct patient types and drive the final results.

Researchers embarking on a claim analysis project will of course usually start with a defined cohort relevant to their research question. Analysis plans and study registrations may require *a priori* cohort definitions using diagnoses and procedure codes. The HSR Definition Builder can fit into this process, either before or after registration, and gives users a way to inspect their definitions on a cohort all receiving the service of interest, therefore providing a view of what the definition may be missing. The counts from the HSR Definition Builder are not precise enough to replace the actual results of a study, and are only built on a sample of claims.

There are limitations and workarounds inherent in this application. The goal was to provide summary counts of overlapping important diagnosis codes in service claims. This creates a large table with small cell counts, which typically cannot be shared or downloaded from a secure data environment for privacy reasons. We represent these masked counts with a range of possible values. Users can increase the sample size (here we used 5,000 claims as a maximum) to have fewer masked cell counts, although this will increase the SAS run time.

We also implemented a number of steps in the SAS program to achieve a manageable number of diagnoses codes to inspect. This included truncating diagnosis codes at four characters, a beneficial feature of the chapter structure of the ICD coding system. This does mean, however, that if analysts want to use more precise codes in their definitions they will have to further inspect the claims data. Another limitation is some low frequency diagnoses codes that are not selected as an important diagnosis code as seen in Table 3. This is likely an issue for testing, like urinalysis, that are recorded in claims with a wide range of diagnosis codes, compared to procedures. We also restricted our claims data to services provided to patients aged 65 and over. If users have a different population in their study, they will have to edit this step in the SAS code, in order to find diagnosis codes relevant to their population.

The current implementation also only provides a snapshot of diagnosis codes recorded in the claim with the service. Users may require a definition that includes previously recorded diagnoses over a set period. Future work may include altering the program to include lookback flags for diagnoses as well. However, even in cases where a lookback is required for a definition, it is still useful to inspect the current diagnoses recorded with the investigated service.

We have found that there is not a clear pathway for analysts to inspect and complete validity checks on cohort definitions. Options include spot checking, where claims are randomly selected in the included and excluded cohorts. Another option is inspecting frequency tables of all diagnosis codes or principal diagnosis codes. In our experience we have found that it is not sufficient to find important codes possibly recorded as secondary diagnoses or codes that have low frequency.

The HSR Definition Builder is an open source program that researchers can use to inspect their proposed cohort definitions. We hope that this application is useful in saving researchers' time when developing cohort definitions, and is a step towards research transparency.

## Author Contributions

**Conceptualization:** Kelsey Chalmers, Valérie Gopinath, Adam G. Elshaug.

**Data curation:** Kelsey Chalmers.

**Formal analysis:** Kelsey Chalmers.

**Funding acquisition:** Adam G. Elshaug.

**Investigation:** Kelsey Chalmers.

**Methodology:** Kelsey Chalmers, Valérie Gopinath.

**Project administration:** Kelsey Chalmers, Adam G. Elshaug.

**Software:** Kelsey Chalmers.

**Supervision:** Valérie Gopinath, Adam G. Elshaug.

**Validation:** Kelsey Chalmers.

**Visualization:** Kelsey Chalmers.

**Writing – original draft:** Kelsey Chalmers.

**Writing – review & editing:** Valérie Gopinath, Adam G. Elshaug.

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
