## [Decision Letter · Decision Letter 0]

24 Aug 2022

PONE-D-22-07582Health service research definition builder: An R Shiny application for exploring diagnosis codes associated with services reported in routinely collected health dataPLOS ONE

Dear Dr. Chalmers,

Thank you for submitting your manuscript to PLOS ONE. After careful consideration, we feel that it has merit but does not fully meet PLOS ONE’s publication criteria as it currently stands. Therefore, we invite you to submit a revised version of the manuscript that addresses the points raised during the review process. Please revise.

We look forward to receiving your revised manuscript.

Kind regards,

Academic Editor

PLOS ONE

https://journals.plos.org/plosone/s/file?id=ba62/PLOSOne_formatting_sample_title_authors_affiliations.pdf".

2. You indicated that ethical approval was not necessary for your study. We understand that the framework for ethical oversight requirements for studies of this type may differ depending on the setting and we would appreciate some further clarification regarding your research. Could you please provide further details on why your study is exempt from the need for approval and confirmation from your institutional review board or research ethics committee (e.g., in the form of a letter or email correspondence) that ethics review was not necessary for this study? Please include a copy of the correspondence as an ""Other"" file."""

NT (Straive) 15 Mar 22: consent information not present. At PRTC, please send back with the following note.

Please provide additional details regarding participant consent. In the ethics statement in the Methods and online submission information, please ensure that you have specified what type you obtained (for instance, written or verbal, and if verbal, how it was documented and witnessed). If your study included minors, state whether you obtained consent from parents or guardians. If the need for consent was waived by the ethics committee, please include this information.

5. Thank you for stating the following in the Funding Section of your manuscript:

“This research was funded by an Arnold Ventures LLC research funding grant.**”**

“The authors received funding from Arnold Ventures LLC for this project (ID 2004079) (https://www.arnoldventures.org/). The funders had no role in study design, data collection and analysis, decision to publish, or preparation of the manuscript.”

6. Thank you for stating the following in the Competing Interests section:

“I have read the journal's policy and the authors of this manuscript have the following competing interests: AE reports receiving personal fees from the Australian state government health departments – Victoria, Queensland, South Australia, as well as the Australian Department of Veterans Affairs, Medibank Ltd, Private Healthcare Australia, and the Australian Defence Force Joint Health Command and grants from the National Health and Medical Research Council outside the submitted work. AE is also on the board of the New South Wales Bureau of Health Information.”

7. In your Data Availability statement, you have not specified where the minimal data set underlying the results described in your manuscript can be found. PLOS defines a study's minimal data set as the underlying data used to reach the conclusions drawn in the manuscript and any additional data required to replicate the reported study findings in their entirety. All PLOS journals require that the minimal data set be made fully available. For more information about our data policy, please see http://journals.plos.org/plosone/s/data-availability.

8. Please include your full ethics statement in the ‘Methods’ section of your manuscript file. In your statement, please include the full name of the IRB or ethics committee who approved or waived your study, as well as whether or not you obtained informed written or verbal consent. If consent was waived for your study, please include this information in your statement as well.

Reviewers' comments:

Reviewer's Responses to Questions

**Comments to the Author**

1. Is the manuscript technically sound, and do the data support the conclusions?

Reviewer #1: Partly

Reviewer #2: Yes

Reviewer #3: Yes

2. Has the statistical analysis been performed appropriately and rigorously? 

Reviewer #1: No

Reviewer #2: N/A

Reviewer #3: Yes

3. Have the authors made all data underlying the findings in their manuscript fully available?

Reviewer #1: No

Reviewer #2: Yes

Reviewer #3: No

4. Is the manuscript presented in an intelligible fashion and written in standard English?

Reviewer #1: Yes

Reviewer #2: Yes

Reviewer #3: Yes

5. Review Comments to the Author

Reviewer #1: This paper supports the process of gathering relevant codes associated with a cohort of patients that is initially pre-defined in some manner.

Overall this problem has been addressed repeatedly in the literature. This paper uses a SAS procedure, HPSPLIT, at its core to address the problem. This is indeed a useful function in SAS, but obviously has been invented already and the paper describes a handful of workflow approaches to using the procedure and uses R (rather than the SAS built in visualizations) for showing the results.

The problem to be addressed is put in a very restrictive context and those who are not familiar with using claims data would have trouble seeing the value, would recommend in future papers that the perspective be broadened.

There are many arbitrary decisions to accommodate technical specifics, such as using only the first 4 digits of ICD codes which are not semantically justified.

The reporting of only outcome statistics for the specific samples used for training is a serious limitation

Reviewer #2: I think that the tool is not easy to use.

One issue is the development of a SAS to R workflow.

I suggest to include a default SAS output to upload when a user opens the app.

The documentation should be richer with more details.

Reviewer #3: I thank the authors for investigating such an important subject; I have the following comments:

- The structure of the manuscript was not organized properly, and it was hard to follow. The Materials and Methods section should be before the Results section.

- The data described that patients younger than 65 were excluded from the study, so how can the results be generalized to people of all ages?

6. PLOS authors have the option to publish the peer review history of their article (what does this mean?). If published, this will include your full peer review and any attached files.

Reviewer #1: No

Reviewer #2: No

Reviewer #3: No

---

## [Author Response · Author response to Decision Letter 0]

8 Nov 2022

Reviewer 1

Thank you to Reviewer 1 for their time and comments on our paper. 

> Overall this problem has been addressed repeatedly in the literature. This paper uses a SAS procedure, HPSPLIT, at its core to address the problem. This is indeed a useful function in SAS, but obviously has been invented already and the paper describes a handful of workflow approaches to using the procedure and uses R (rather than the SAS built in visualizations) for showing the results.

In this paper, we describe our methods and developed tools for building thorough health service indicators. This is useful and important for other researchers developing service measures (whether related to low-value care or otherwise). We have not come across similar approaches or tools described in the literature. The need for such a solution has become clearer to us after our own internal audits on published indicators showed frequent or important diagnosis codes in claims data had sometimes not been included. The benefit of using R is due to the shiny package, which is not available in SAS, and creating a dashboard that can be used by non-coders and those who do not have direct access to claims data. 

> The problem to be addressed is put in a very restrictive context and those who are not familiar with using claims data would have trouble seeing the value, would recommend in future papers that the perspective be broadened.

We agree with Reviewer 1 that this - low-value service indicators - is a somewhat niche problem. A large number of health service research studies, however, do use administrative data and create cohorts based on service and diagnosis codes (as mentioned in our introduction). We have written our paper and built our tool for a research audience, where we assume at least one collaborator is familiar with claims data and the processes involved in creating these initial indicators/cohorts. 

> There are many arbitrary decisions to accommodate technical specifics, such as using only the first 4 digits of ICD codes which are not semantically justified.

This is a challenge that we came across while developing this process, so thank you to Reviewer 1 for raising this issue. Ultimately this is a process and tool for data exploration and visualization. Like many visualization approaches, decisions around summarizing data have to be made. A simple example is choosing the number of bins to use to display data in a histogram. 

We decided on using a limit of four characters for ICD diagnosis codes because of the organization of diagnosis information in the ICD codes. For example, M17.1 describes “Unilateral primary osteoarthritis”, M17.11 describes “Unilateral primary osteoarthritis, right knee” and M17.12 describes “Unilateral primary osteoarthritis, left knee”. The actual informative condition for the purpose of this exercise is unilateral primary osteoarthritis, which just requires four characters of the ICD codes. 

> The reporting of only outcome statistics for the specific samples used for training is a serious limitation

Thank you to Reviewer 3 for pointing this out. We had originally decided to report only on the original training set since our goal was to select the diagnosis codes ‘important’ to this sample. After revisiting the paper, we decided to report the sensitivity, specificity and misclassification rates on a validation sample of claims in Table 2 (and have made updates to the results on page 9 to reference these results). 

Reviewer 2

Thank you to Reviewer 2 for their time and comments on our manuscript. 

> I think that the tool is not easy to use. One issue is the development of a SAS to R workflow. I suggest to include a default SAS output to upload when a user opens the app.

We have revisited the R tool and package and have taken the opportunity to make a number of improvements, including Reviewer 2’s useful suggestion to have an option to upload an example data set. Our other improvements include: 

- Including checkboxes for Include/Exclude options for codes

- Adding a row ID column so sorting is easier

- Updating the label input as a type box/drop down menu so previous labels can be saved

- Adding colors to the action buttons

> The documentation should be richer with more details.

As well as the changes made above, we have included more instructions on the application when the user opens it. All of these updates can be observed in the Github repository. 

Reviewer 3

> I thank the authors for investigating such an important subject.

We thank Reviewer 3 for their time and thoughts on our manuscript. 

> The structure of the manuscript was not organized properly, and it was hard to follow. The Materials and Methods section should be before the Results section.

We had originally organized in this way to follow some previous PLOS ONE publications that we had found useful in understanding the purpose/outputs (the results) prior to the details of the Materials and Methods. We do take Reviewer 3’s point, however, and have updated the text so the Materials and Methods are before the Results. 

> The data described that patients younger than 65 were excluded from the study, so how can the results be generalized to people of all ages?

Our goal was to build indicators for services provided to people 65 years and older, and never to generalize to data on people of all ages. Users working with other populations will have to change this age restriction. We have now noted this in the discussion section (p18, lines 1-3): 

“We also restricted our claims data to services provided to patients aged 65 and over. If users have a different population in their study, they will have to edit this step in the SAS code, in order to find diagnosis codes relevant to their population.”

---

## [Decision Letter · Decision Letter 1]

22 Dec 2022

Health service research definition builder: An R Shiny application for exploring diagnosis codes associated with services reported in routinely collected health data

PONE-D-22-07582R1

Dear Dr. Chalmers,

We’re pleased to inform you that your manuscript has been judged scientifically suitable for publication and will be formally accepted for publication once it meets all outstanding technical requirements.

Kind regards,

Academic Editor

PLOS ONE

Additional Editor Comments (optional):

Reviewers' comments:

Reviewer's Responses to Questions

**Comments to the Author**

1. If the authors have adequately addressed your comments raised in a previous round of review and you feel that this manuscript is now acceptable for publication, you may indicate that here to bypass the “Comments to the Author” section, enter your conflict of interest statement in the “Confidential to Editor” section, and submit your "Accept" recommendation.

Reviewer #3: All comments have been addressed

Reviewer #4: All comments have been addressed

2. Is the manuscript technically sound, and do the data support the conclusions?

Reviewer #3: Yes

Reviewer #4: Yes

3. Has the statistical analysis been performed appropriately and rigorously? 

Reviewer #3: Yes

Reviewer #4: Yes

4. Have the authors made all data underlying the findings in their manuscript fully available?

Reviewer #3: No

Reviewer #4: Yes

5. Is the manuscript presented in an intelligible fashion and written in standard English?

Reviewer #3: Yes

Reviewer #4: No

6. Review Comments to the Author

Reviewer #3: (No Response)

Reviewer #4: (No Response)

7. PLOS authors have the option to publish the peer review history of their article (what does this mean?). If published, this will include your full peer review and any attached files.

Reviewer #3: No

Reviewer #4: No

---

## [Editor Report · Acceptance letter]

4 Jan 2023

PONE-D-22-07582R1 

Health service research definition builder: An R Shiny application for exploring diagnosis codes associated with services reported in routinely collected health data 

Dear Dr. Chalmers:

I'm pleased to inform you that your manuscript has been deemed suitable for publication in PLOS ONE. Congratulations! Your manuscript is now with our production department. 

Kind regards, 

on behalf of

Dr. Robert Jeenchen Chen 

Academic Editor

PLOS ONE